# Alternative Utilization of Vegetable Crop: Pumpkin Polysaccharide Extract and Their Efficacy on Skin Hydration

**Setinee Chanpirom** [1,2], **Nisakorn Saewan** [1,3] **and Tawanun Sripisut** [1,2,*]

1   School of Cosmetic Science, Mae Fah Luang University, Chiang Rai 57100, Thailand
2   Cosmetic for Beauty and Wellness Research Unit, Mae Fah Luang University, Chiang Rai 57100, Thailand
3   Cosmetic and Beauty Innovations for Sustainable Development (CBIS) Research Group,
     Mae Fah Luang University, 333, Moo.1, Thasud, Muang, Chiang Rai 57100, Thailand
*   Correspondence: tawanun.sri@mfu.ac.th; Tel.: +66-53-916-833

**Abstract:** Traditional pumpkin (*Cucurbita moschata*) and Japanese pumpkin (*C. maxima*) consist of natural polysaccharides. From a scientific basis, natural polysaccharides could be applied to improve hydration in the cosmetic field. The purified polysaccharide was extracted and deproteinized with the $CaCl_2$ method. Japanese pumpkin showed the higher value of physicochemical properties including yield ($12.96 \pm 0.60\%$), total polysaccharide content ($0.89 \pm 0.04$ mg/mL), swelling capacity ($4.00 \pm 0.00\%$), swelling index ($1.04 \pm 0.00\%$), solubility ($126.67 \pm 5.77\%$), viscosity ($1.25 \pm 0.00$ cps), water capacity ($0.93 \pm 0.15$ g/g) and oil absorption capacity ($5.93 \pm 0.06$ g/g) than traditional pumpkin. Additionally, Japanese pumpkin ($IC_{50}$ $9.30 \pm 0.58$ µg/mL) provided higher antioxidant activity by DPPH assay than traditional pumpkin ($IC_{50}$ $9.98 \pm 0.25$ µg/mL). The evaluation of efficacy on skin hydration in fifteen Thai volunteers indicated that Japanese pumpkin showed non-skin irritation. An extract concentration of 0.05–0.1% showed a significantly increased effect in moisturizing ability according to concentration ($p < 0.05$). This result supported that it was safe and effective to use as a moisturizer for cosmetic products.

**Keywords:** pumpkin; *Cucurbita* sp.; polysaccharide; antioxidant; skin hydration

## 1. Introduction

Polysaccharides are naturally derived polymer molecules which are originated from simple sugars [1] and can be sources of energy or structural material for the biological functions of cells [2]. They are obtained from various natural sources such as chitosan from the exoskeleton of crustaceans, algae and fungi [3], dextran from fungi and bacteria of the *Leuconostoc* genus [4], pectin from plant cell walls [5], and carrageenan from red seaweeds [6]. Polysaccharides have diverse properties because of the components of their monosaccharides, length of chains, the conformation of these chains, and their bond to other constituents [7]. Natural polysaccharides have been reported to demonstrate biological activities. Ceylon spinach, rose mallow and cotton rose polysaccharides [8,9] can be used as active biopolymers to improve a skin moisturizing effect in cosmetic formulations. In addition, polysaccharides from brown seaweed, cucumber, semen cassia, and pumpkin have been reported to have antioxidant activities which are non-toxic to the skin [10–12]. Glycyrrhiza polysaccharide displayed antimicrobial activity [7]. Sulfated polysaccharides derived from seaweed and red algae also possess many activities such as anticoagulant, antiviral, immune-inflammatory, antilipidemic, and antioxidant activities [7,13,14].

Pumpkin is an annual creeping plant belonging to the Cucurbitaceae family which is widely used as a vegetable [2,15]. The main important nutrients in pumpkin are polysaccharides, carotenoids, mineral elements, and amino acids [10]. Many components of pumpkin show biological functions; antioxidant, antimicrobial, antidiabetic activities, and hypoglycemic effects [10,15]. Traditional and Japanese pumpkins are popular species that are used as food in Thailand. Traditional pumpkin (*Cucurbita moschata* Duchesne) is a summer

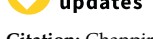



squash that has a rough peel whereas Japanese pumpkin (*C. maxima* Duchesne) is a thick winter squash with a hard rind, and smooth peel (Figure 1). These two pumpkins have been widely used as food sources of nutrients, especially polysaccharides. Pumpkin polysaccharide is characterized based on extraction and deproteinization in this study. Moreover, the purity of polysaccharides and its total monosaccharide content are also important [16].

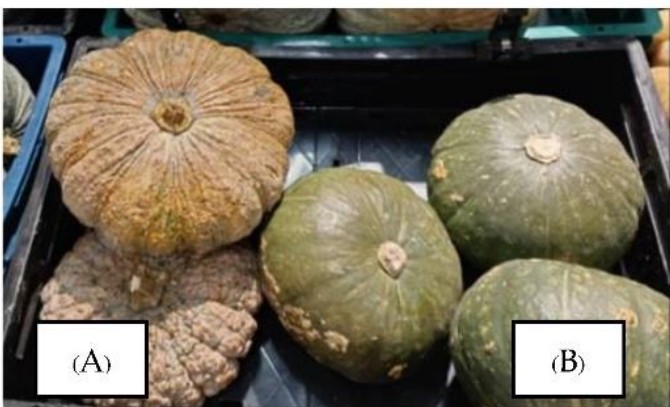

**Figure 1.** Traditional pumpkin (*Cucurbita moschata* Duchesne) (**A**) and Japanese pumpkin (*C. maxima* Duchesne) (**B**).

Over the last decade, polysaccharides derived from plants were used as natural active ingredients for cosmetic and pharmaceutical industries [17,18]. The standardization of pumpkin polysaccharides is necessary for industrial use. The aim of this study was to standardize and compare the physicochemical properties, and antioxidant activity of traditional and Japanese pumpkin polysaccharides. In addition, clinical efficacy of skin hydration using Tewameter® TM 300 and Corneometer® CM was investigated on the study volunteers over a short period of time.

## 2. Materials and Methods

### 2.1. Plant Materials

Fresh fruits of traditional and Japanese pumpkins were purchased from Makro supermarket in Chiang Rai, Thailand, December 2018.

### 2.2. Chemical Materials

We purchased 99.9% of absolute ethanol (AR grade), and diethyl ether (AR grade) from Fisher scientific, Waltham, MA, USA. Sodium lauryl sulfate was obtained from BASF (Thai) Ltd., Bangkok, Thailand. Olive oil, and hyaluronic acid were purchased from Namsiang Co., Ltd., Bangkok, Thailand. Additionally, 2,2-Diphenyl-1-picrylhydrazyl (DPPH), calcium chloride, hydrochloric acid, phenol, sulfuric acid, and sodium hydroxide pellets (AR grade) were purchased from Sigma-Aldrich Corporation, St. Louis, MO, USA. L-ascorbic acid and Glucose were obtained from Chem-Supply Pty Ltd., Gillman, SA, Australia.

### 2.3. Extraction of Crude Pumpkins Polysaccharide

The fresh fruit of each pumpkin was cut and dried in a hot air oven at 45 °C for 3 days. Small pieces of pumpkin were grounded into powder form. The water-soluble pumpkin polysaccharide was obtained from 200 g of pumpkin powder by extracting with 2 L of distilled water boiled at 95 °C for 2 h. It was cooled to room temperature before it was filtered [10]. The supernatant liquid was collected and centrifuged at 4500 rpm for 20 min. The pumpkin polysaccharide was precipitated by adding 200 mL of absolute ethanol at 4 °C for 1 week based on method described by Zhang, et al. [19], with slight modification. The absolute ethanol was evaporated before drying the precipitate in a hot air oven at 45 °C for 48 h [20]. The dried precipitate was washed with absolute ethanol and diethyl ether, respectively, using the method described by Chen and Huang with a slight modification [2].

*2.4. Deproteinization by CaCl$_2$ Method*

The dried precipitate of the crude pumpkin polysaccharide was adjusted to pH 9–10 with 2% NaOH solution and heated to 85 °C. After that, 5% *w/v* of CaCl$_2$ solution was added to the sample. Then, it was cooled and filtered. The filtrate was adjusted to pH 7 with 20% HCl solution. Then 200 mL of absolute ethanol was added to the pumpkin polysaccharide precipitate. The absolute ethanol was removed by an evaporator under reduced pressure before drying the precipitate in a hot air oven at 45 °C for 48 h. The dried precipitate was washed with absolute ethanol and diethyl ether, respectively [10].

*2.5. Analysis of Pumpkin Polysaccharide*

2.5.1. FT-IR Characterization of Pumpkin Polysaccharide

The dried extract was made into a pellet with KBr powder. FT-IR (Perkin Elmer Spectrum-GX FT-IR spectrometer) of the sample was recorded with the transmittance mode from 4000–400 cm$^{-1}$ and 16 scans [9].

2.5.2. $^{13}$C NMR Spectrum Analysis of Pumpkin Polysaccharide

The $^1$H and $^{13}$C NMR spectra were recorded on 500 and 125 MHz Bruker spectrometers in D$_2$O using TMS as an internal standard.

2.5.3. Total Polysaccharide Content

Total polysaccharide content was determined with a phenol-sulfuric acid method using glucose as the standard [9]. First, 100 μL of the extract solution was mixed with 150 μL of concentrated sulfuric acid (96–98%) and 30 μL of 5% phenol was added. The mixture was heated at 95 °C for 10 min before it was added to a 96-well microplate to measure its absorbance at 490 nm [21].

2.5.4. Solubility Test

The solubility of the pumpkin polysaccharide was determined by pouring 10 mg of the sample into 10 mL of distilled water. Then, it was saturated overnight. The supernatant liquid was transferred into an evaporating dish, where it was completely evaporated dry. After that, the dried residue was weighed [9]. United States pharmacopeia solubility criteria was shown in Table 1.

**Table 1.** United States Pharmacopeia solubility criteria.

| Descriptive Term | Parts of Solvent Required for 1 Part of Solute |
|---|---|
| Very soluble | Less than 1 |
| Freely soluble | From 1–10 |
| Soluble | From 10–30 |
| Sparingly soluble | From 30–100 |
| Slightly soluble | From 100–1000 |
| Very slightly soluble | From 1000–10,000 |
| Practically insoluble, or insoluble | Greater than or equal to 10,000 |

2.5.5. Viscosity Test

First, 0.3 g of the sample was weighed and saturated in 7.5 mL of distilled water and saturated overnight. The mixture was adjusted to 10 mL, and mixed thoroughly. Its viscosity was measured by using a viscometer (Brookfield DVII+ Pro, Middleborough, MA, USA) under the ambient condition (spindle no.21, 20 rpm) [9].

2.5.6. Swelling Index and Swelling Capacity

First, 0.1 g of the dry sample was mixed with 0.1 mL of absolute ethanol and 25 mL of distilled water, respectively based on British pharmacopoeia, with slight modification. A vortex mixer (Vortex Daihan Scientific, Wonju, Korea) was used for 2 min. After that, the

mixture was left for 10 min and centrifuged (Hettich/Universal 320R, Tuttlingen, Germany) at 1000 rpm for 10 min [22].

### 2.5.7. In Vitro Water Absorption and Oil Absorption Capacity

In vitro water absorption was determined by mixing 0.1 g of pumpkin polysaccharide with 6 mL of distilled water in a centrifuge tube for 18 h. The mixture was centrifuged at 3000 rpm for 20 min and the residue weight was recorded. In addition, the oil absorption capability of the samples was determined using same procedure but it was mixed with olive oil [9].

### 2.5.8. Determination of Antioxidant Activity by DPPH Assay

Antioxidant activity of the samples was determined by using DPPH assay, 100 µL of DPPH in ethanol solution ($6 \times 10^{-5}$ M) was added to 100 µL of the sample or standard and dissolved in water. Various concentrations of L-ascorbic acid solution (1–8 µg/mL) were used as standard in which the calibration curve and percent of inhibition were plotted. The concentrations of the sample were replaced with 2–10 µg/mL. The mixture was incubated in the dark at room temperature for 30 min and measured at 517 nm [23].

### 2.6. Skin Hydration Efficacy

The study performed on the human volunteers was delineated by the Declaration of Helsinki. The volunteers were Thai women aged between 20 to 25 years old. They were informed about the study both verbally and in written form. They all signed a written consent form, which was approved by the ethical committee of the Mae Fah Luang University (REH-62068) before participating in the study.

### 2.6.1. Skin Irritation Test

The polysaccharide solution was prepared (0.05 and 0.10% *w/v*) in distilled water for use in a single application closed patch test. Water was used as a negative control while sodium lauryl sulfate (SLS) was the positive control. It was observed for 24 h, followed by the removal of Fin chamber (8 mm, SmartPractice, Phoenix, AZ, USA). Irritation was determined by the Mean Irritation Index (M.I.I) [9]. The classification of M.I.I is shown in Table 2.

**Table 2.** The classification of mean irritation index.

| M.I.I | Class |
|---|---|
| M.I.I < 0.20 | Non-Irritation (NI) |
| 0.20 < M.I.I < 0.50 | Slightly Irritating (SI) |
| 0.50 < M.I.I < 1 | Moderately Irritating (MI) |
| M.I.I > 1 | Irritating (I) |

### 2.6.2. In Vivo Skin Hydration Efficacy

Moisturizers, body lotion, and sunscreen were not applied on the volunteers' body and the tested area for 12 h before the study. All volunteers rested in a room maintained at 25 °C and 40–60% relative humidity for 15 min before their hydrated skin was monitored using Tewameter® TM 300 and Corneometer® CM 825 at the inner forearm. The test solution was prepared by dissolving the polysaccharide extract (0.05 and 1.0% *w/v*), and hyaluronic acid (0.05 and 0.1% *w/v*) in water. The skin hydration level was recorded before and after applying the samples test solution for 15, 30, 45, 60, 90, 120, 150, 180, and 210 min [9]. Skin hydration efficacy was calculated using Equation (1)

$$\text{Skin hydrating efficacy (\%)} = [(A_t - A_0)/A_0] \times 100 \qquad (1)$$

where $A_t$ is skin capacitance at a specified time. $A_0$ is skin capacitance at baseline.

### 2.7. Statistical Analysis

All experiments were performed in triplicate. The SPSS software ver. 22.0 was used for statistical analysis. Independent t-test and one-way ANOVA were used for evaluating the efficacy of the samples. *p* values < 0.05 were regarded as statistically significant. The significance was set at a reliability of 95% and the clinical evaluation was expressed as mean $\pm$ SEM.

## 3. Results and Discussion

### 3.1. Preparation of Pumpkin Polysaccharides

Pumpkin was extracted as water-soluble polysaccharides in hot water and deproteinized by $CaCl_2$ based on the modified protocol of Chen et al. [10]. Additionally, diethyl ether was used to wash the precipitates. $CaCl_2$ method was used to remove proteins from the crude extract and a high amount of purified polysaccharide was obtained for analysis. The physical appearance of the purified traditional pumpkin polysaccharide (P-TP) appeared as white to pale yellow powder, whereas the purified Japanese pumpkin polysaccharide (P-JP) was a brownish-yellow powder. Percent yields of P-TP and P-JP were $1.04 \pm 0.26$ and $12.96 \pm 0.60\%$, respectively. This result showed that P-JP had a higher yield than P-TP (Table 3).

**Table 3.** Comparison of extraction and deproteinization of pumpkin polysaccharides.

| Pumpkin | P-TP | P-JP |
|---|---|---|
| Crude Polysaccharide (g) | $3.93 \pm 0.59$ [a] | $32.42 \pm 0.99$ [b] |
| Protein (g) | $1.45 \pm 0.52$ [a] | $4.21 \pm 0.08$ [b] |
| Yield of protein (%) | $0.73 \pm 0.16$ [a] | $2.10 \pm 0.04$ [b] |
| Purified polysaccharide (g) | $2.07 \pm 0.33$ [a] | $25.91 \pm 1.20$ [b] |
| Yield of purified polysaccharide (%) | $1.04 \pm 0.26$ [a] | $12.96 \pm 0.60$ [b] |

Results are expressed as means $\pm$ SD. Different lowercase indicates a significant difference ($p < 0.05$).

### 3.2. Physicochemical Properties of Pumpkin Polysaccharide

In the solubility test, the samples were dissolved in distilled water and their residues were measured after drying. Polysaccharide has a strong affinity to water molecules due to the presence of OH group. P-TP was sparingly soluble in water with a clear solution while P-JP was slightly soluble with a light brown solution (Table 1). The extract having total polysaccharide content was examined by phenol-sulfuric acid method. Concentrated sulfuric acid broke down polysaccharides to monosaccharides and then reacted with phenol to produce a yellow to orange color. The high intensity of the color indicated a high amount of monosaccharide content. From the result in Table 4, P-JP ($0.89 \pm 0.04$ mg/mL) was significantly greater than P-TP ($0.17 \pm 0.00$ mg/mL). It may have a higher moisturizing effect on the skin than the traditional one. The swelling capacity and swelling index determined the volume change of both extracts upon inundation with water. The difference in the swelling capacity of both samples was associated with the abilities of polysaccharides to absorb moisture and hold water; polysaccharide matrix can hold water that impacts the swelling and viscosity of the solution [8]. Both samples had low swelling capacity due to their less viscous solution and low viscosity. The ability of polysaccharide to absorb water depends on the content of its hydrophilic side chain [9]. Additionally, both samples showed low ability to absorb water, indicating their ability to absorb water was poor [24]. Moreover, oil absorption is controlled by hydrophobic or the backbone regions of polysaccharides due mainly to the ability of polysaccharides to physically entrap oil. This indicates that non-polar sites of protein chains can absorb oil [24]. Both samples showed a higher ability to absorb oil than water. Regarding their deproteinization, both samples showed a low ability to absorb oil due to remaining low amount of protein [25].

**Table 4.** Physicochemical properties and antioxidant activity of pumpkin polysaccharides.

| Parameters | P-TP | P-JP |
|---|---|---|
| Total polysaccharide content (mg/mL) | 0.17 ± 0.00 [a] | 0.89 ± 0.04 [b] |
| Swelling Index (mL/mL) | 1.02 ± 0.00 [a] | 1.04 ± 0.00 [a] |
| Swelling capacity (%) | 2.00 ± 0.00 [a] | 4.00 ± 0.00 [a] |
| Solubility in water (%) | 100.00 ± 0.00 [a] | 126.67 ± 5.77 [b] |
| Viscosity (cps) | 1.00 ± 0.00 [a] | 1.25 ± 0.00 [a] |
| Water absorption (g/g) | 0.53 ± 0.06 [a] | 0.93 ± 0.15 [b] |
| Oil absorption (g/g) | 4.80 ± 0.36 [a] | 5.93 ± 0.06 [b] |
| Antioxidant activity ($IC_{50}$, µg/mL) | 9.98 ± 0.25 [a] | 9.30 ± 0.58 [a] |

Results are expressed as means ± SD. Different lowercase indicates a significant difference ($p < 0.05$).

### 3.3. Antioxidant Properties of Pumpkin Polysaccharide Using DPPH Assay

DPPH radical scavenging activity is widely used to determine the antioxidant activity of pumpkin polysaccharide by eliminating free radicals. In the process, the purple color of radicals turns into pale yellow color [26]. Polysaccharide is a polymer consisting of a group linked by a glycosidic bond and directly donates electrons to free radicals to terminate radical reaction [23]. From previous reports, cucurbit polysaccharide was shown to have good scavenging effect for use as a potential antioxidant [16]. Its antioxidant activity was determined by $IC_{50}$. The inhibition activity of P-TP and P-JP increased with increased concentration. $IC_{50}$ is defined as the concentration of a substance that inhibits 50% of DPPH activity. It was found that 9.30 ± 0.58 µg/mL of P-JP and 9.98 ± 0.25 µg/mL of P-TP can inhibit 50% radicals, which are comparable to L-ascorbic acid 2.73 ± 0.48 µg/mL (Table 4). From the above information, Japanese pumpkin polysaccharide showed better properties than traditional pumpkin polysaccharide, therefore it was selected for further study.

### 3.4. FT-IR Characterization of Polysaccharide

The IR-spectra of the purified Japanese pumpkin polysaccharide (P-JP) is shown in Figure 2. The absorption bands of P-JP at 3366 cm$^{-1}$ (O-H stretch), 2935 cm$^{-1}$ (C-H stretch), 1590 cm$^{-1}$ (N-H bend) were determined as a residual protein, and 1453 cm$^{-1}$ (C-H bend), 1273 cm$^{-1}$ (C-O stretch), 1050 cm$^{-1}$ (C-O stretch), 858–995 cm$^{-1}$ (C-H) were assigned to the glycosidic bond [2,11].

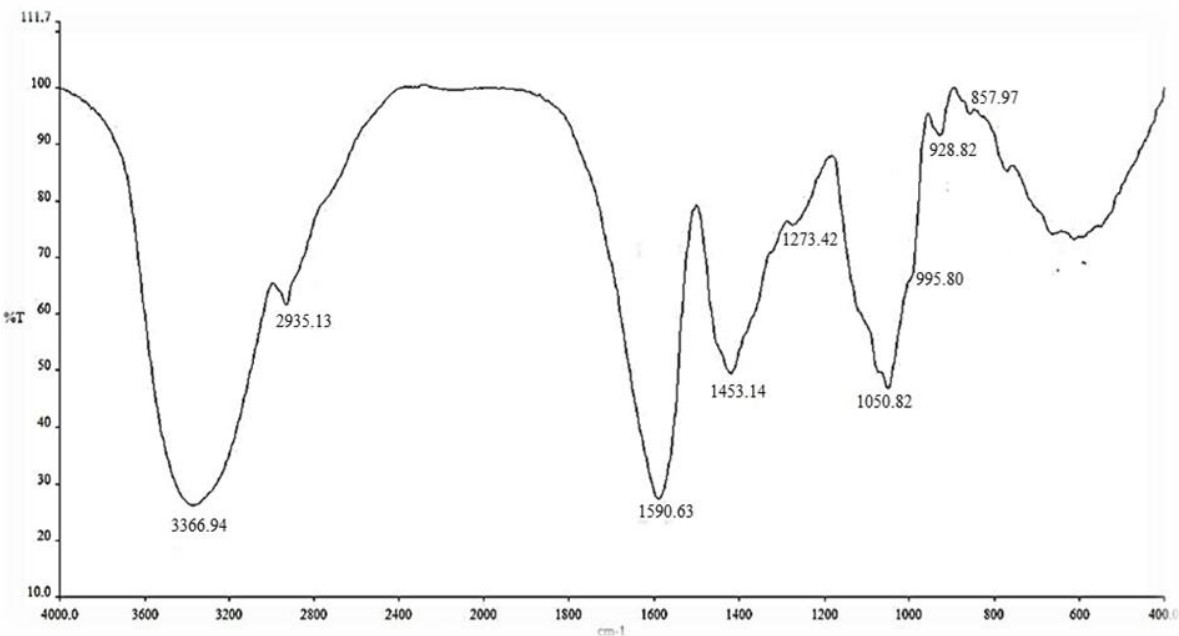

**Figure 2.** FTIR spectra (in KBr) of the purified Japanese (P-JP) pumpkin polysaccharide.

### 3.5. $^1$H and $^{13}$C NMR Spectra Analysis of Pumpkin Polysaccharide

The $^1$H and $^{13}$C NMR spectra of P-JP are shown in Figure 3. The $^1$H NMR of P-JP presented the signal of the glycosidic ring at around 3.40 to 4.20 ppm [27]. The high field signals (1.25–1.40 ppm) were assigned to methyl protons at C6 [2]. The $^{13}$C NMR spectrum of P-JP displayed anomeric carbon signals from 92.2 to 103.7 ppm and multiple non-anomeric carbon signals in the abroad region from 60.1 to 81.4 ppm. In addition, the low field signal at around 171.2 to 182.7 ppm was assigned to the carboxyl carbon [28].

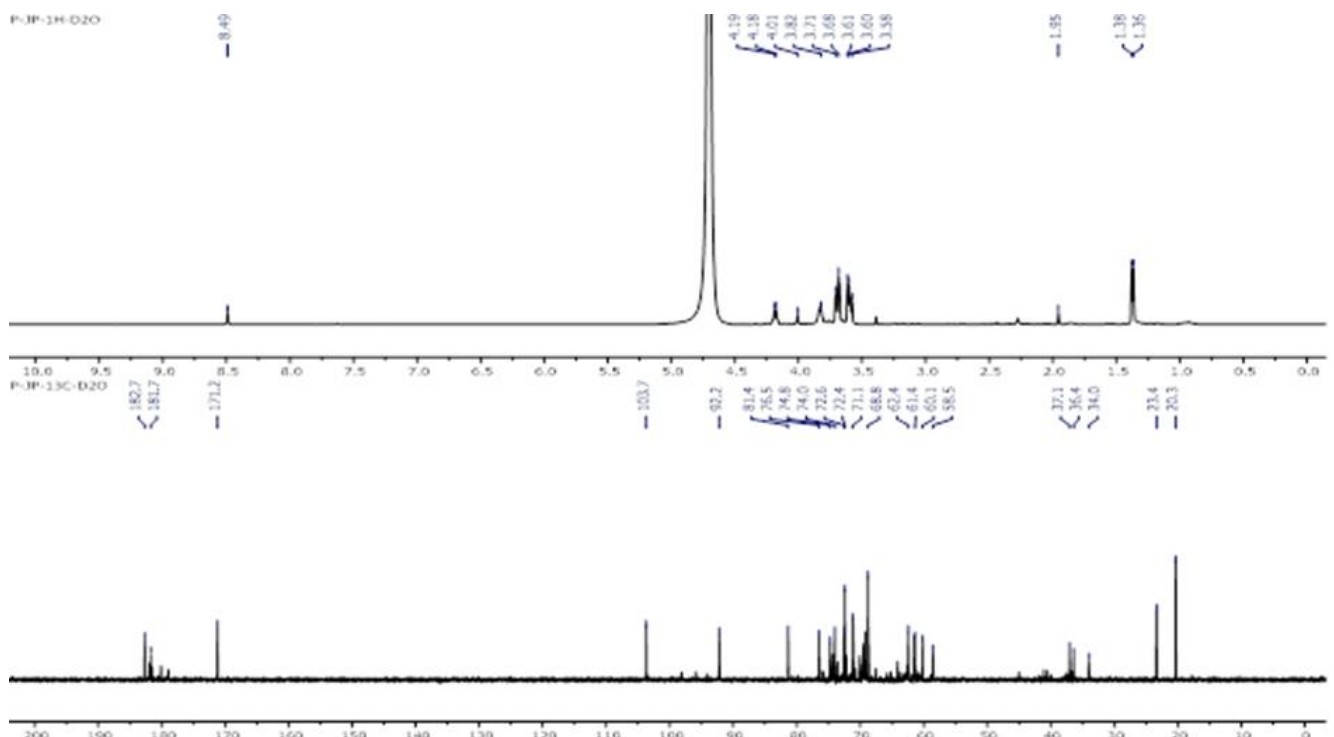

**Figure 3.** $^1$H (D$_2$O, 500 MHz) and $^{13}$C NMR spectra (D$_2$O, 125 MHz) of P-JP.

### 3.6. Skin Hydration Efficacy

3.6.1. Skin Irritation Test

The efficacy of P-JP was evaluated for the skin irritation test using a closed patch test on 15 female volunteers. The test was classified on The Mean Irritation Index (M.I.I) according to the range of values in Table 2. The result showed that at 0.05–0.1% *w/v* P-JP had an M.I.I value equal to 0. This shows it did not irritate the skin. So, it was classified as a cosmetic ingredient that is safe to use.

3.6.2. In Vivo Skin Hydration Efficacy

Polysaccharide, mainly from natural sources, is widely used as a natural moisturizer in the cosmetic industry. From the skin irritation test, pumpkin polysaccharide is shown to be safe and its moisturizing effect could be further tested on the volunteers' skin. Transepidermal water loss (TEWL) and the skin hydrating efficacy were evaluated by Tewameter® TM 300 and Corneometer® CM 825, respectively. In this study, two concentrations (0.05 and 1.0% *w/v*) of P-JP were evaluated and compared to hyaluronic acid. From previous reports, the molecular weight of pumpkin polysaccharide was about 23 kDa and it was classified as low molecular weight whereas hyaluronic acid was in the range of 20–300 kDa. The compounds that were higher than 500 Daltons, formed a film on the skin to reduce transdermal water loss and indirectly improved skin hydration [29,30].

At 0.1% *w/v* concentration, hyaluronic acid showed higher skin hydrating efficacy than P-JP (Figures 4 and 5). Furthermore, the result from Tewameter® TM 300 (Figure 4)

showed TEWL in the volunteers' skin; a high value of TEWL was considered a high amount of water loss to the environment. TEWL values compared to the baseline were reduced. While over time TEWL significantly increased consequently ($p < 0.05$). All samples could maintain the value for 210 min. The relationship between hydration and water loss showed good efficacy. Pumpkin polysaccharide enhanced moisturizing ($p < 0.05$) compared to the baseline. With increase in time, the water loss value increased, which reduced the skin hydration ($p < 0.05$). Therefore, the pumpkin polysaccharide can maintain moisture for a short period of time. Polysaccharides compared with 0.05 and 0.1% *w/v* of hyaluronic acid were evaluated for TEWL at 0–210 min. There was significant difference ($p < 0.05$) observed between both. There is increased water loss when the epidermal barrier is disrupted, whereas a lower TEWL suggests that the skin barrier function is maintained [31]. Therefore, P-JP showed slight occlusive effects, preventing the skin's moisture from evaporating in short-term experiments. In Figure 5, skin hydration efficacy (%) of all the samples was compared with the control (0 min). Even 0.1% of hyaluronic acid was the highest skin hydration efficacy (%), 0.05% *w/v* of P-JP and hyaluronic acid were similar. Both concentrations of P-JP retained skin hydration better than the control. Short-term skin hydration efficacy lasted for 210 min. Therefore, 0.05–0.1% P-JP prolonged skin hydration to 210 min in a single application.

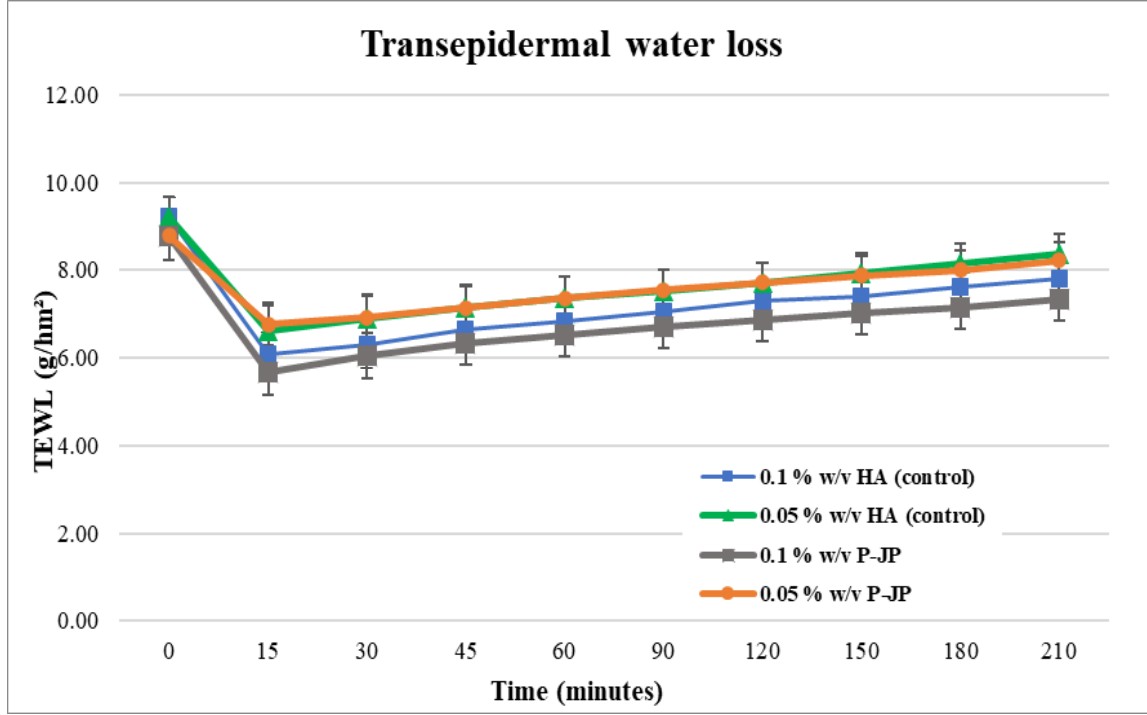

**Figure 4.** Transepidermal water loss of 0.05 and 0.1% *w/v* of the purified Japanese pumpkin polysaccharides (P-JP) in comparison with 0.05 and 0.1% *w/v* of hyaluronic acid (HA) evaluated for TEWL from 0–210 min.

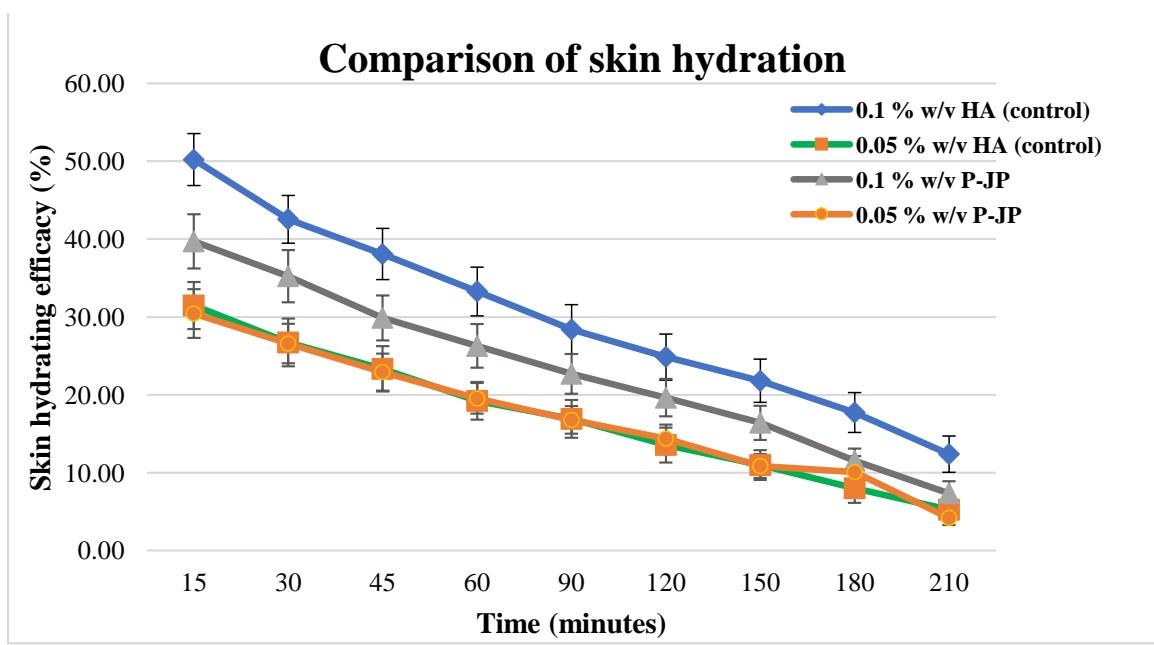

**Figure 5.** Skin hydrating efficacy of 0.05 and 0.1% $w/v$ of the purified Japanese pumpkin polysaccharides (P-JP) in comparison with 0.05 and 0.1% $w/v$ of hyaluronic acid (HA) evaluated for skin hydration from 0–210 min.

## 4. Conclusions

Pumpkin is a vegetable crop which is rich in vitamins, minerals, antioxidant compounds, and polysaccharide. The pumpkin polysaccharide was extracted by hot water and deproteinized. The purified Japanese pumpkin polysaccharides (P-JP) showed higher polysaccharide and physiochemical properties than the purified traditional pumpkin polysaccharide (P-TP). Interestingly, both pumpkin polysaccharides showed good antioxidant activity. Additionally, P-JP showed a potent skin hydrating efficacy. From this information, P-JP was recommended to be a safe and effective moisturizer to use in topical products for skin dryness and anti-aging treatment.

**Author Contributions:** Conceptualization, T.S.; formal analysis and investigation, S.C. and T.S.; writing—original draft preparation, S.C., N.S. and T.S.; writing—review and editing, S.C., N.S. and T.S.; funding acquisition, N.S. and T.S. All authors have read and agreed to the published version of the manuscript.

**Funding:** This research was financially supported by Mae Fah Luang University.

**Institutional Review Board Statement:** The study was conducted according to the guidelines of the Declaration of Helsinki and approved by the Ethics Committee of Mae Fah Luang University. The certificate of analysis (COA) and protocol numbers are 068/2562 and REH-62068, respectively.

**Informed Consent Statement:** Informed consent was obtained from all the volunteers involved in the study.

**Data Availability Statement:** Data are contained within the article.

**Acknowledgments:** The authors would like to thank Mae Fah Luang University for funding and laboratory facilities. We are grateful to Surat Laphookhieo, School of Science, Mae Fah Luang University, for recording NMR spectrum.

**Conflicts of Interest:** The authors declare no conflict of interest in this study. There was no role for any contributions in this study from a grant funder.

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
