# Peer review of "Alternative Utilization of Vegetable Crop: Pumpkin Polysaccharide Extract and Their Efficacy on Skin Hydration"

_cosmetics, doi:10.3390/cosmetics9060113_

Round 1

Reviewer 1 Report

Dear authors,

Congratulations on your manuscript.

I only have few suggestions:

There is no statistical analysis on Figure 4 and Figure 5, please provide them;

The Introduction section can be improved with more references related to the tested species and a cleared Objective;

The conclusion can be more concise and should have no refferences.

Author Response

Reviewer

Comment

MS (cosmetics-1998394)

Reviewer 1

The introduction section can be improved with more references related to the tested species and a cleared objective.

The sentence was revised in the Introduction and Objective parts

Statistical analysis on Figures 4 and Figure 5.

p value was added in Statistical analysis.

The sentence was revised.

The conclusion can be more concise and should have no references.

The sentence was revised.

Reviewer 2 Report

This is an interesting manuscript on a well conducted study

i only have two major concerns 

1) English language needs improvement 

2) show p value in tables 3 and 4, write 

test used for analysis under the table

Author Response

Responses to Reviewer Comments and Suggestions

Reviewer

Comment

MS (cosmetics-1998394)

Reviewer 1

The introduction section can be improved with more references related to the tested species and a cleared objective.

The sentence was revised in the Introduction and Objective parts

Statistical analysis on Figures 4 and Figure 5.

p value was added in Statistical analysis.

The sentence was revised.

The conclusion can be more concise and should have no references.

The sentence was revised.

Reviewer 2

English language needs improvement.

English was checked and proved by Global Proofreading.

Show p value in tables 3 and 4, write

test used for analysis under the table.

p value was added in Statistical analysis.

Also, it presented under Tables 3 and 4.

-

-

The references were rearranged in all MS

-

-

The title was reworded from “……their skin hydration efficacy” to “…….their efficacy on skin hydration
